# Deprescribing as an Opportunity to Facilitate Patient-Centered Care: A Qualitative Study of General Practitioners and Pharmacists in Japan

**DOI:** 10.3390/ijerph20043543

**Published:** 2023-02-17

**Authors:** Kenya Ie, Reiko Machino, Steven M. Albert, Shiori Tomita, Mio Kushibuchi, Masanori Hirose, Takahide Matsuda, Chiaki Okuse, Yoshiyuki Ohira

**Affiliations:** 1Department of General Internal Medicine, St. Marianna University School of Medicine, Kawasaki 214-8525, Japan; 2Department of General Internal Medicine, Kawasaki Municipal Tama Hospital, Kawasaki 214-8525, Japan; 3Department of Behavioral and Community Health Sciences, Graduate School of Public Health, University of Pittsburgh, Pittsburgh, PA 15621, USA; 4Tama Family Clinic, Kawasaki 214-0013, Japan; 5Division of Clinical Epidemiology, Jikei University School of Medicine, Tokyo 105-8461, Japan

**Keywords:** polypharmacy, deprescribing, potentially inappropriate medication

## Abstract

Deprescribing has recently been applied to address polypharmacy, particularly among older adults. However, the characteristics of deprescribing that are likely to improve health outcomes have not been well studied. This study explored the experiences and perspectives of general practitioners and pharmacists with regard to deprescribing in older adults with multimorbidity. A qualitative study was conducted involving eight semi-structured focus group interviews with 35 physicians and pharmacists from hospitals, clinics, and community pharmacies. Thematic analysis was applied to identify themes using the theory of planned behavior as a guide. The results illustrated a metacognitive process, as well as influencing factors, through which healthcare providers commit to shared decision making for deprescribing. Healthcare providers acted on the basis of their attitudes and beliefs on deprescribing, the influence of subjective norms, and perceived behavioral control for deprescribing. These processes are influenced by factors such as drug class, prescribers, patients, deprescribing experience, and environment/education. Healthcare providers’ attitudes, beliefs, and behavioral control (along with deprescribing strategies) evolve in a dynamic interplay with experience, environment, and education. Our results can serve as a foundation for the development of effective patient-centered deprescribing to improve the safety of pharmaceutical care for older adults.

## 1. Introduction

An increasing number of older adults have multiple chronic diseases, also known as multimorbidity [1,2]. This trend has led to increases in polypharmacy [3]. Polypharmacy patients are at higher risk of adverse health outcomes, potentially due to adverse drug events, nonadherence as a result of complex drug regimens, and poor patient care coordination [4]. With this background, deprescribing, defined as “the process of tapering or stopping drugs, aimed at minimizing polypharmacy and improving outcomes [5]”, has been increasingly recognized as a critical issue when caring for older adults.

The evidence on the effects of deprescribing on clinical outcomes among older adults remains controversial [6,7,8], potentially owing to the ambiguity of both polypharmacy patient populations and deprescribing interventions. However, it has been suggested that patient-centered multimodal deprescribing, compared to uniform deprescribing without elements of shared decision making (e.g., reducing a specific risky medication), may be more effective in improving clinical outcomes [9]. Therefore, it is essential to explore the shared decision-making process that leads to clinically effective deprescribing.

Previous studies on patient perspectives on polypharmacy revealed insufficient patient awareness of the potential seriousness of adverse drug events [10,11]. In a survey conducted in a tertiary healthcare facility, two-thirds of patients were satisfied with their current medications, and one-third were reluctant to stop taking medicines [11]. Conversely, another study reported that more than 90% of a cohort of mostly older adults were willing to stop a medication if their prescriber deemed it feasible [12]. Thus, elucidating the perspectives of healthcare providers on prescribing is essential for the development of clinically effective interventions. However, relatively little research has explored the attitudes and beliefs of healthcare professionals regarding deprescribing [5,13,14,15,16,17], and evidence is particularly lacking in the general medicine setting.

This study aimed to qualitatively explore the experiences and perspectives of general practitioners and pharmacists on deprescribing in multimorbid older adults, with the goal of developing effective patient-centered deprescribing strategies.

## 2. Materials and Methods

### 2.1. Study Design

A qualitative study design was implemented using semi-structured focus group interviews. The Consolidated Criteria for Reporting Qualitative Research (COREQ) were used to report study methods, study context, findings, analysis, and interpretations [18]. The details of the responses to each of the COREQ-32 items are reported in Appendix A.

### 2.2. Data Collection

From June 2022 to December 2022, physicians and pharmacists with experience in deprescribing were invited from two community hospitals, four primary care clinics, and one community pharmacy. Written consent was obtained from all participants. A purposeful sampling technique was applied to represent key stakeholders.

Focus group interviews (FGI) were semi-structured and based on a framework developed by a group discussion of three researchers (K.I., S.A., and R.M.) after a review of the literature, including the theory of planned behavior (TPB) [19]. TPB is a cognitive theory that explains how an individual’s specific behavior depends upon their intention to engage in that behavior. Intentions are influenced by three variables: attitude, subjective norms, and perceived behavioral control. TPB has been applied to a variety of behaviors, and recent publications have successfully used TPB to investigate barriers and facilitators to prescribing by healthcare providers [20,21]. We sought to apply TPB for conceptualization in this study because it highlights a person’s intention to engage in a specific behavior at a specific time and place.

All questions were reviewed and rewritten if the items were ambiguous or unnecessary. The following three items were included in the semi-structured questions:Experiences with deprescribing interventions and perceived challenges and opportunities for deprescribing;Perceived tips for effective patient-centered deprescribing;Thoughts on innovations to improve patient engagement that can be applied in clinical practice.

### 2.3. Analysis

Thematic analysis was applied to identify themes, as indicated by the saliency, frequency, and elaboration of the codes. The codes were derived using a hybrid deductive-inductive approach following the approach used by Hahn et al. [10]. The transcripts were independently read and coded by each researcher (K.I. and R.M.) to identify themes and subthemes. The initial codebook was collaboratively developed by K.I. and R.M. through discussion after the first FGI. For subsequent FGIs, K.I. and R.M. independently performed the coding and conceptualization and checked the codebook agreement after each round of the FGI. The proportion of codes with discrepancies among researchers was 17.6%. These discrepancies were recorded and resolved through group discussions.

Along with the deductive derivation of codes, we simultaneously used our a priori research hypothesis and TPB to capture emergent themes in the transcripts. Data were analyzed after each round of the focus group to develop preliminary codes to identify new concepts. A codebook with a detailed definition of each code was used for categorization and conceptualization. The interviews were iteratively continued until the researchers agreed on theoretical saturation.

## 3. Results

Eight FGIs were conducted. Of the 35 respondents, 19 (54.3%) were general practitioners and 16 (45.7%) were clinical pharmacists. The mean duration of clinical experience of the respondents was 11.5 years (SD 8.2, range 1–38 years). The practice settings of respondents were the clinic for 16 (45.7%), the hospital for 14 (40%), and the community pharmacy for five (14.3%). Participant characteristics are shown in Table 1.

Overall, participants were proactive in deprescribing. Participants often viewed polypharmacy as a problem and looked for opportunities to deprescribe. Some participants had well-planned deprescribing strategies, although their comfort levels varied. The conceptual model of the metacognitive process of prescribing and the various levels of facilitators and barriers in daily practice is illustrated in Figure 1.

### 3.1. Theme 1: Contextual Supports and Barriers

The participants described the influence of various contexts on deprescribing. Factors that facilitated deprescribing included a work setting that ensured care continuity (capacity to treat the same patients across multiple visits), multidisciplinary collaboration, and partnership with local and in-hospital pharmacists. Hospital-based providers also viewed inpatient settings as advantageous for deprescribing. Advantages include the allocation of time for intervention, the availability of postintervention follow-up, the delegation of prescribing authority to the inpatient team, and the fact that some patients viewed hospital admission as an opportunity to adjust medication treatment. The contexts that inhibit deprescribing include a lack of incentives, loss of information due to fragmented care, and competition with outpatient settings for time and information.

### 3.2. Theme 2: Attitude and Belief toward Deprescribing

When healthcare providers identified a patient’s prescription as a problem, an important factor in this context was the healthcare provider’s attitudes and beliefs about polypharmacy and deprescribing when initiating a conversation about deprescribing.

#### 3.2.1. Sub-Theme 2.1: Positive Attitude toward Deprescribing

The positive attitude of the healthcare provider toward deprescribing was a crucial element in initiating dialogue. This was the key to overcoming the various barriers discussed in this study.

“*Everyone here is eager to deprescribe*.”(A-32)

#### 3.2.2. Sub-Theme 2.2: Attitude of Strategic Compassion toward Long-Term Deprescribing

Many participants expressed an attitude of “strategic compassion” along with positive attitudes toward deprescribing. Participants preferred to wait for the opportunity to deprescribe while building an ongoing relationship with patients who were prescribed potentially inappropriate medications (PIMs) but were not ready to reduce/stop their medications.

“*Really, it takes time, so I don’t think it will work if you rush. I think it would be better to build a good relationship first and if we could taper medications at multiple visits, that would be great*.”(C-20)

“*I’ve had a few patients who, for whatever reason, eventually accepted tapering medications when I stopped pushing it*.”(H-12)

These attitudes and beliefs were nurtured through treatment of patients with multimorbidity and polypharmacy. In addition, participants described the importance of education, the environment, and culture in achieving this expertise.

#### 3.2.3. Sub-Theme 2.3: Experiences That Shaped Attitudes toward Deprescribing

Some participants experienced a change in their fixed notions about the necessity of prescribing through empathy with the patient and the patient’s perspectives on prescribed medications.

“*She really wanted to continue taking benzodiazepines just because that made her sleep better. Through the conversation with her, I came to reconsider whether reducing them really makes her happy*.”(H-15)

#### 3.2.4. Sub-Theme 2.4: Environment/Education That Shaped Attitudes toward Deprescribing

When polypharmacy is introduced in daily conversations in clinical settings, clinicians naturally become aware of the problem and learn how to address it, including tacit knowledge across the group.

“*I think it is part of the culture of our clinic. Everybody cares about medication appropriateness and health maintenance. There was a problem named ‘polypharmacy’ in a patient’s chart taken over from my colleague. So, I thought, ‘Oh, this is a problem’ and started to take that seriously*.”(G-44)

“*When I was having trouble with a patient who didn’t accept my deprescribing proposal, I was asked by my preceptor, ‘Is it really a problem? Who is affected by that?’ I was surprised and thought that was true*.”(H-10)

### 3.3. Theme 3: Subjective Norm Related to Polypharmacy

Some participants felt pressure from society or their peers to address polypharmacy and potentially inappropriate medications, while others did not seem to be as concerned.

“*When I worked with young doctors, one of them said, ‘Why are you prescribing so many medicines?’ Since then, this makes me feel pressured to reduce prescriptions*.”(G-16)

“*I feel that the idea, ‘less is better’, is ingrained in me. Quite often I feel guilty for not being able to taper medications*.”(H-16)

Although the subjective norm and/or peer pressure can suppress polypharmacy, participants expressed concern that these pressures may lead prescribers to push too hard for deprescribing, potentially resulting in doctor–patient communication errors.

### 3.4. Theme 4: Perceived Behavioral Control about Deprescribing

In addition to internal factors, such as attitudes and beliefs described in Theme 2, there were a variety of external factors that influenced behavioral control when initiating conversations with patients about deprescribing.

#### 3.4.1. Sub-Theme 4.1: Drug Factors Related to Perceived Behavioral Control

In the presence of the following factors, healthcare providers stated that they would feel comfortable deprescribing, possibly due to increased perceived behavioral control over these drugs: low adherence, drugs without clear indication, drugs without clear patient expectations, concerns about adverse drug events, numerical goals achieved (e.g., HbA1c goal), symptomatic drugs with limited potential benefits, symptomatic drugs without persistent symptoms, and overlapping drug effects/interactions/contraindications.

In contrast, barriers to control included drugs with an unknown reason, drugs that have been taken for a long time without any adverse events, drugs beyond the control of the interventionist, severity of conditions that can relapse with deprescribing, drugs requiring tapering, and adequate polypharmacy with multiple chronic conditions.

#### 3.4.2. Sub-Theme 4.2: Prescribing Physician Factors Related to Perceived Behavioral Control

This subtheme is applicable when the usual prescribing physician and interventionist are different, such as in a hospital setting. Facilitating factors included a collaborative interventionist–treating physician relationship and the idea of strengthening this relationship through deprescribing. This theme is best illustrated by the following quote:

“*I am taking deprescribing as an opportunity to deepen our relationships with other medical institutions*.”(G-4)

Barriers included clinical inertia (resistance to changing the status quo), indifference of prescribers to polypharmacy, lack of openness to opinions from other professionals, lack of partnership or opportunities for collaboration, and established PIM patterns for each prescriber. The degree of behavioral control appears to be altered depending on the external factors described above. This degree of behavioral control, together with the attitudes, beliefs, and subjective norms of the participants regarding polypharmacy, was considered to promote or inhibit their intention to start conversations about prescribing with their patients.

### 3.5. Theme 5: Factors Influencing Intention to Talk about Deprescribing

This theme describes factors that influence whether healthcare professionals initiate conversations about deprescribing, with a focus on safe prescribing. A stronger intention reflects a greater likelihood of starting a conversation with the goal of deprescribing.

#### 3.5.1. Sub-Theme 5.1: Readiness for An Opportunity to Deprescribing

Participants emphasized the importance of readiness to talk about deprescribing. This is particularly important in an outpatient setting because of time constraints and multiple agendas during each visit.

“*I am ready for an opportunity to deprescribe, especially when the topic happens to come up. Incorporating deprescribing into my outpatient clinic, that’s what I have recently tried to do*.”(G-10)

#### 3.5.2. Sub-Theme 5.2: Patient Factors Related to Intention to Talk about Deprescribing

Several physicians and pharmacists were aware of the common characteristics of the patients, which led to stronger intentions to discuss deprescribing. These included the patient’s wish to deprescribe, a good understanding of one’s disease, a trusting relationship, proactive patients, difficulty in taking drugs due to worsening conditions, a change in patient awareness of life events or health status, and a change in patient awareness due to adverse events.

“*These conversations make me understand what is important for them. Sometimes, I feel that the doctor–patient relationship has deepened a bit, although the medications have not been reduced*.”(H-20)

Patient-level barriers included patient/family trust in their prescribing physician, expectations for the drug, resistance to changing the status quo, experience of worsening symptoms with deprescribing, less active treatment, excessive reliance on healthcare providers’ decisions, and lack of a patient–physician partnership in prescribing.

“*The patient’s fear of losing the trust they have established with the prescribing physician is quite significant and often a barrier to reducing regimens*.”(C-22)

These facilitators and barriers at the patient level, together with the external factors described in Theme 4, are summarized in Table 2.

### 3.6. Theme 6: Deprescribing Strategies

When deciding to start a shared decision-making (SDM) process for deprescribing, some participants had well-planned deprescribing strategies. These strategies were based on the experience of participants and, in some cases, their training (Table 3).

#### 3.6.1. Sub-Theme 6.1: Patient/Family Participation in Deprescribing SDM

Most of the participants emphasized good communications and tried to involve patients and their families in conversations about prescribing. Mutual understanding and shared decision making between the interventionist and the patient were highly valued.

#### 3.6.2. Sub-Theme 6.2: Providing Adequate Medical Information to Patients

Physicians and pharmacists also emphasized the importance of conveying appropriate medical information during the shared decision-making processes. Participants explained the disadvantages of polypharmacy and the benefits of deprescribing, recent drug transitions, alternative medications, societal views on polypharmacy, and the “latest” drug information.

#### 3.6.3. Sub-Theme 6.3: Applying Motivational Interviewing Strategies

This sub-theme was mainly emphasized by physicians, probably because of their familiarity with behavioral approaches. Participants reported that they utilized basic motivational interview skills with the intention of increasing the patient’s proactiveness and understanding of their medication treatment. Reassurances that discontinued medications can be resumed if necessary and introducing nonpharmacological therapy were also emphasized during the deprescribing conversation.

#### 3.6.4. Sub-Theme 6.4: The Mindset for Successful Deprescribing

In addition to “strategic compassion toward long-term deprescribing” (Sub-Theme 2.2), participants mentioned a related strategy of waiting for patients to become more ready to accept deprescribing. This change in readiness is often triggered by changes in health status or unintended information from social networks, rather than paternalistic recommendations from healthcare professionals.

## 4. Discussion

This study illustrates the metacognitive process through which healthcare providers commit to shared decision making regarding deprescribing. In our conceptualization, attitudes, beliefs, and behavioral control over prescribing emerge from experience, environment, and education along specific pathways.

Our results contribute to the body of evidence on barriers and facilitators of deprescribing. In particular, many of the identified deprescribing barriers were consistent with those reported in previous studies [5,13,14,15,16,17]. Contextual barriers consistent with previous reports include a lack of incentives and information due to fragmented care [14,16,17], especially in outpatient settings. Potential prescriber barriers include clinical inertia, lack of awareness and openness regarding deprescribing, and lack of partnership or opportunities for collaboration, which have also been reported previously [5,14,17]. In addition to these barriers, we identified several drug-related facilitators, which are considered indicators of deprescribing, including low adherence, drugs without clear indication, no specific patient expectations, concerns about adverse drug events, numerical goals achieved, symptomatic drugs with limited potential benefits or no persistent symptoms, and overlapping drug effects/interactions/contraindications [5,13].

The first and most explicit contribution of our study to the growing qualitative literature is the use of TPB [19] as a background theory to explore the experiences and perspectives of general practitioners and pharmacists on deprescribing. Our evolving themes could be well organized along the TPB elements of attitudes, subjective norms, and perceived behavioral control over the intention to commit to deprescribing. The results showed that healthcare providers seek to deprescribe on the basis of a positive attitude toward deprescribing and sometimes on the basis of pressure from a subjective norm. Their perceived behavioral control to commit to a shared decision-making process for deprescribing, along with their attitude toward deprescribing, was largely influenced by a variety of factors, including drug factors, prescriber factors, the experience of deprescribing, and the environment/education that shapes attitudes toward deprescribing. Attitudes, subjective norms, and perceived behavioral control affect the intent to commit to deprescribing, although these factors may or may not be considered consciously. Since shared decision making on deprescribing is a preference-sensitive decision, as suggested in a previous study [18], patient-specific barriers and facilitators largely influenced the intention of healthcare providers to deprescribe.

Another novelty of our study lies in the conceptualization of the feedback process in which attitudes, beliefs, and degree of perceived behavioral control over deprescribing are not static but are rather continually updated according to a healthcare professional’s experience, education, and the environment. For example, contrary to previous reports that described a social norm of “doctors heal diseases with drugs” as a barrier to deprescribing [22], our participants emphasized the pressure of subjective norms, which sometimes led them to address polypharmacy. Some recognized that their experience with patients, education, and the environment gave them a more flexible attitude, the attitude of strategic compassion toward long-term deprescribing, resulting in more successful deprescribing. Participants also noticed that a mutual understanding process between the patient and the healthcare provider sometimes led to more active patient involvement in the process.

Compared to previous studies regarding physicians’ and/or pharmacists’ views on deprescribing, our results are unique in that participants overall were optimistic about dealing with patients with polypharmacy, whereas many previous qualitative studies emphasized challenges and difficulties in tackling problematic polypharmacy [20,23]. In our sample, some participants viewed deprescribing as an opportunity to strengthen doctor–patient relationships and relationships with other healthcare providers. Elements for effective prescribing can be embedded in clinical training, which is not necessarily specific to polypharmacy. Further studies are required to confirm this hypothesis.

Our study had several limitations. First, there may have been a risk of bias with a purposeful sampling design. However, it was necessary to recruit participants from facilities where healthcare providers were actively involved in deprescribing while ensuring maximum possible diversity. Social desirability bias with a group interview design was another limitation. Although we were thorough in protecting personal information and reassured participants that there were no right or wrong answers, we could not rule out the possibility that some statements may not accurately depict these perspectives.

## 5. Conclusions

In this study, we conceptualized the metacognitive process of deprescribing and related influencing factors using TPB. Healthcare providers acted on the basis of their attitudes and beliefs on deprescribing, the influence of subjective norms, and perceived behavioral control over deprescribing. These processes are influenced by factors such as drugs, prescribers, patients, deprescribing experience, and environment/education. Furthermore, our results highlight the process of nurturing attitudes, beliefs, and behavioral control over prescribing by reflecting on feedback from personal experience, the environment, and education. These results can serve as a foundation for the development of effective patient-centered deprescribing to improve the quality and safety of care for older adults.

## Figures and Tables

**Figure 1 ijerph-20-03543-f001:**
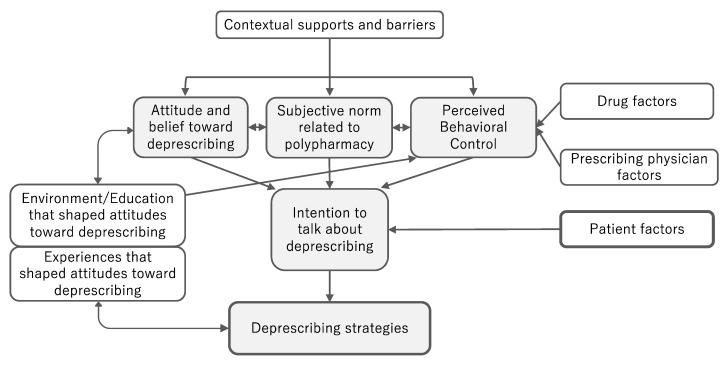
The metacognitive process of deprescribing and influencing factors among general practitioners and pharmacists.

**Table 1 ijerph-20-03543-t001:** Participant characteristics (N = 35).

Characteristics	Number (%)
Sex	
Female	15 (42.9)
Male	20 (57.1)
Academic degree	
M.D.	19 (54.3)
Pharm. D.	16 (45.7)
Years in clinical practice	
1–5	8 (22.9)
6–10	13 (37.1)
11–20	10 (28.6)
21+	4 (11.4)
Main practice setting	
Primary care clinic	16 (45.7)
Hospital	14 (40.0)
Community pharmacy	5 (14.3)
Practice location	
Urban	19 (54.3)
Suburban	4 (11.4)
Rural	12 (34.3)

**Table 2 ijerph-20-03543-t002:** Facilitators and barriers affecting healthcare professional’s perception and behavior.

Sub-Theme		Codes
Drug factors related to perceived behavioral control about deprescribing	Facilitators	Low adherence
		Concerns for adverse drug events
		Numerical goals achieved (e.g., HbA1c goal)
		Drugs with limited potential benefits
		Prophylactic drugs with little expected benefit
		Symptomatic drugs without persistent symptoms
		Overlapping effects/interactions/contraindications
		Drugs without any expectations from the patient
	Barriers	Unclear reason for prescribing
		Drugs taken for a long time without any adverse events
		Drugs beyond the control of the interventionist
		Drugs without numerical indicator
		Drugs for which efficacy is difficult to determine
		Severity of conditions that can relapse with deprescribing
		Drugs that require tapering
		Drugs that are dispensed in single packets
		Hesitation of financial waste due to discontinuation of dispensed drugs
		Treatment with a combination of multiple mechanisms of action
		Adequate polypharmacy for multiple chronic conditions
Prescribing physician factors related to perceived behavioral control about deprescribing	Facilitators	Collaborative interventionist–prescribing physician relationship
		Intent to strengthen doctor–patient relationship through deprescribing
		Intent to strengthen the relationship with other HCPs
	Barriers	Clinical inertia (resistance to changing the status quo)
		Prescribers’ indifference to polypharmacy
		Lack of openness to opinions from other professions
		Lack of partnership or opportunities for collaboration
		Insufficient relationships between the intervention team, prescribing physician, and community pharmacies
		Established PIMs patterns for each prescriber
Patient factors related to intention to talk about deprescribing	Facilitators	Patient’s wish to reduce medication
		Adequate understanding of the disease by the patient
		Well-established physician–patient relationship
		Patient preference for active participation in care decision making
		Experience of improved physical condition due to deprescribing
		No deterioration of symptoms after previous deprescribing
		Difficulty swallowing pills due to deterioration of health condition
		Experience of adverse drug events
		Health orientation due to lifestyle changes and awareness of health issues
	Barriers	Patient and family expectations for the effect of medicines
		Patient/family trust in their prescribing physicians
		Resistance to changing the current condition
		Less preference for active participation in care decision-making
		Lack of patient–physician partnership for healthcare decisions
		Experience of worsening symptoms with deprescribing

HCP: healthcare professional; PIM: potentially inappropriate medication.

**Table 3 ijerph-20-03543-t003:** Deprescribing strategies suggested by general practitioners and pharmacists.

Sub-Theme	Codes and Representative Quotations
Patient/family participation in deprescribing SDM	Mutual understanding and shared decision making through dialogue “*It may be time-consuming, but listening to how they think on the basis of face-to-face conversation would be the most important part in involving them (A–4).*” “*I think we (patient and doctor) are trying to align our perceptions of the specific medicine that he or she is taking. If the balance is tilted in the direction of quitting after sufficient dialogue, then, well, we will do it (E-36).*”
	Involving the patient’s family in conversations about deprescribing “*It’s also helpful to involve family members in the conversation and talk about reducing medication. If a family member says,* ‘*Mom, you don’t have to take this anymore,’ it is often easier for the patient to accept it (A-47).*”
Providing adequate medical information to patients	Assurance from health care providers that deprescribing is feasible. “*Although it was quite challenging for us to reduce her diabetes meds, she agreed to reduce them when the director said* ‘*Your diabetes is well under control. It’s almost cured (H-5)!’*”
	Emphasizing the benefits of deprescribing *Less adverse events (A-49), reduced financial burden (A-48), ease of medication management (B-47)*
	Explanation of harms related to polypharmacy and PIM “*I usually try to explain so that they can be aware of the influence of polypharmacy. For example, kidney function declines with age so it may not be as safe as it used to be (E-44).*”
	Proposing conditions of exchange for deprescribing “*Sleep problems also involve exercise, lifestyle, and diet, as well as medications, and the various problems that patients have do not get better with just one solution (H-23).*”
Applying motivational interviewing strategies	Attempt to deprescribing through motivational interviewing “*Ask them how they feel about it, then dig down into both the positive and the negative aspects, and find a right topic that the patient might be motivated (G-24).*”
	Improving patient proactiveness by helping them to understand therapeutic options “*She was taking the medicine simply because her doctor prescribed it, but when she understood the reasons for the prescription, the patient became more proactive in her treatment (E-26).*”
	Reassurance that discontinued medications can be resumed if necessary “*I often reassure them that they can resume taking the medicines if symptoms recur (A-42).*”
The mindset for successful deprescribing	Attitude of strategic compassion toward long-term deprescribing “*Really, it takes time, so I don’t think it will work if you rush. I think it would be better to build a good relationship first, and if we can taper medications across multiple visits, which would be great (C-20).*”
	Taking advantage of serendipity “*When the topic of a drug just happens to appear in a magazine, there are cases where this can be an opportunity to start conversation about reducing the drug (G-34).*”

SDM: shared decision making, PIM: potentially inappropriate medication.

## Data Availability

The data used and analyzed during the current study are available from the corresponding author upon reasonable request.

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
