# Peer review of "Deprescribing as an Opportunity to Facilitate Patient-Centered Care: A Qualitative Study of General Practitioners and Pharmacists in Japan"

_ijerph, 2023, doi:10.3390/ijerph20043543_

Round 1

Reviewer 1 Report

Overall, this work is good; it is important and leads to the positive conclusions stated by the authors.

- The authors should carefully correct spelling and any typo errors in entire manuscript

-It is need to make additional adjustments to the article's title

-I suggest the authors to add the phrases of 'background', 'methods', 'results' and 'conclusion' in Abstract section
A more detailed conclusion is required
-

Author Response

Thank you for your time and effort in reviewing our manuscript. Your suggestions were very helpful and appreciated.

- The authors should carefully correct spelling and any typo errors in entire manuscript
Per your suggestion, one of the coauthors whose mother language is English carefully read through the manuscript and made necessary language editing. The revisions made have been recorded using track changes in the revised manuscript.

-It is need to make additional adjustments to the article's title
Thank you for your comments regarding the article’s title. We revised the title as follows to improve readability and clarity. Please let us know if you have any additional suggestion regarding the article’s title.
“Deprescribing as an opportunity to facilitate patient-centered care: A qualitative study of general practitioners and pharmacists in Japan”

-I suggest the authors to add the phrases of 'background', 'methods', 'results' and 'conclusion' in Abstract section
According to the journal’s Instructions for Authors, it is stated that “the abstract should be a single paragraph and should follow the style of structured abstracts, but without headings”. We will leave it to the editorial decision, but please indicate if you still believe it is necessary to add the headings.

-A more detailed conclusion is required-
We appreciate your suggestions for clarifying the message of this paper. We added a more detailed description in page 11, lines 555-558 to better convey what this study adds to the literature.

Reviewer 2 Report

This manuscript entitled by "Deprescribing as an opportunity to facilitate patient-centered care: A qualitative study from perspectives of general practitioners and pharmacists" by Ie K. et al. indicated a deprescribing for improvement of safety in pharmaceutical care. This manuscript is very important in this field. Bur some corrections may be needed. In introduction section, it was better to light the research question in this manuscript by adding current other researches with references.  In Table 1, it was better to increase number of respondents using statistical power.  

Author Response

Thank you for your constructive feedback regarding our manuscript. Per your suggestion, one of the coauthors whose mother language is English carefully read through the manuscript and made necessary language editing. The revisions made have been recorded using track changes in the revised manuscript.

In introduction section, it was better to light the research question in this manuscript by adding current other researches with references.
Per your suggestion, we added references in the introduction section to better highlight the importance of the research question.

In Table 1, it was better to increase number of respondents using statistical power. 
Thank you for your comment. Because this paper is a qualitative rather than quantitative study, it was not originally intended to ensure statistical power. The number of respondents in this study was determined by checking for theoretical saturation based on adequate triangulation among the researchers. Furthermore, we believe that the number of respondents in this study, 35, is not small considering similar qualitative studies.

Reviewer 3 Report

The study is very interesting and the topic is very important as the authors explained. The methodology section is not absolutely clear. From my side it is very difficult to analyze type of questions included in the study. It is not clear also what general distribtution of the answers is. What exactly means the abbreviation in brackets, for example A-32. 

The subtopics are well organized but the answers explanation is not  detailed. I think that additionally can be explained rating of the answers of healthcare professionals.

Author Response

Thank you for your comments. Because this paper is a qualitative rather than quantitative study, it was designed to collect and analyze non-numerical text data to understand concepts, opinions, or experiences. A detailed explanation of qualitative study design and its reporting guidelines can be found here.
https://www.equator-network.org/?post_type=eq_guidelines&eq_guidelines_study_design=qualitative-research&eq_guidelines_clinical_specialty=0&eq_guidelines_report_section=0&s=

As displayed in the supplemental table, this paper was conducted and written in accordance with the COREQ reporting guidelines for qualitative research.

It is not clear also what general distribtution of the answers is.
The subtopics are well organized but the answers explanation is not detailed. I think that additionally can be explained rating of the answers of healthcare professionals.
Thank you for your question. We have included the actual number and proportion for each stratum to reveal the distribution of participant characteristics. As stated above, since our study is a qualitative study design, we have not counted the number or rating of the answers. In most qualitative studies, the frequency and number of responses are not the focus of the research, but rather the in-depth content and meaning of even single responses. We hope you understand these characteristics of qualitative research, but if you have additional questions or comments we will consider modifications, so please be specific in your suggestions.

What exactly means the abbreviation in brackets, for example A-32.
The abbreviation and numbers displayed in brackets are used to link specific quotations to an anonymized facility code and list of quotations. We used this style with reference to previously published studies such as the following, and believe that it is not an unusual notation in studies using text data from interviews. However, we are open to deleting these abbreviations and numbers if reviewers and the editorial office feel that would be better for the readability.
PLoS ONE 11(4): e0151066.
Clinical Interventions in Aging 2018:13 1401–1408

Round 2

Reviewer 2 Report

This manuscript is corrected accoring to the reviewers comments.